# KCl Extracted from Chlorine Bypass Dust as Activator for Plain Concrete

**DOI:** 10.3390/ma14206091

**Published:** 2021-10-15

**Authors:** Hong-Beom Choi, Jin-Man Kim, Sun-Mi Choi, Sung-Su Kim

**Affiliations:** 1Construction Test & Certification Department, Korea Institute of Civil Engineering and Building Technology, Goyang-si 10223, Korea; hongbeomchoi@kict.re.kr; 2Department of Architectural Engineering, Kongju National University, Cheonan City 31080, Korea; 3Eco-Friendly Concrete Research Center, Kongju National University, Cheonan City 31080, Korea; smchoi@kongju.ac.kr; 4Department of Environmental Energy Engineering, Kyonggi University, Suwon 16227, Korea; sskim@kyonggi.ac.kr

**Keywords:** by-product, chlorine bypass dust, alkali activator, recycling, KCl

## Abstract

This study demonstrated the use of KCl separated from chlorine bypass dust (CBD) as an activator for plain concrete. The separated KCl was mixed with either ground granulated blast-furnace slag (BFS) alone, or a mixture of BFS and cement. The mixed paste of separated KCl and BFS set within 24 h, and exhibited a compressive strength of 22.6 MPa after 28 d. The separated KCl, cement, and BFS mixture exhibited a more rapid setting and a higher initial activity. Further, the compressive strength at 28 d was 57.7 MPa, which was 26.2% higher than that of the mixture without the activator. Water curing of samples with added separated KCl led to the generation of hydrocalumite, or Friedel’s salt. However, this hydrocalumite was decomposed while being cured under autoclave conditions at 180 °C. Overall, KCl was an effective activator for composite materials containing cement, and resulted in superior properties compared to mineral admixtures without an activator.

## 1. Introduction

Advances in concrete-based construction technologies have facilitated rapid industrial development and the construction of cutting-edge residential infrastructures. Concrete is an essential construction material and has cement as its main ingredient. Cement generates carbon dioxide during high-temperature sintering due to chemical decomposition, which leads to the emission of approximately 0.8 tons of carbon dioxide per ton of cement during production [1,2]. Consequently, the carbon dioxide emissions from the cement industry account for 8% of global carbon dioxide emissions [3]. Due to environmental concerns regarding these substantial greenhouse gas (GHG) emissions, as well as the depletion of cement raw material reserves, recent research has focused on reducing the proportion of cement in concrete.

Industrial by-products with hydraulic or pozzolanic reactivity have been widely investigated as substitutes that might allow for the reduction of cement usage, ground granulated blast furnace slag (BFS) and fly ash (FA) being two representative cementitious materials [4,5]. These industrial by-products are also activated by alkali components, where the addition of an alkali activator can lead to enhanced reactions by forcibly destroying the thin film on the surface of industrial by-product particles [6,7]. There are various studies on the properties of mixtures that use strong alkali activators, such as Ca(OH)_2_, NaOH, and KOH, instead of Portland cement [8,9,10,11,12]. Further, enhanced strength has been achieved by adding alkali activators to the cementitious mixtures even when using industrial by-products [13,14,15,16,17]. A number of studies of additive materials have been reported in which the addition of polyvinyl alcohol (PVA) fibers and nano-SiO_2_ improves both the quality and the development direction of existing alkali-activated mortar [18,19,20]. However, alkali-activated cement relies on costly alkali activators, and is thus less favorable in terms of performance and economic efficiency compared to mixtures based on Portland cement. Because these issues must be addressed to encourage the practical use of alkali-activated cement, research on the development of cheaper activators will be needed.

Cement processing provides the optimal environmental solution for the safe treatment of plastic wastes. However, plastics with a high chlorine content, such as polyvinyl chloride (PVC), will increase the chlorine content of the clinker when it burns in a cement kiln. Thus, the clinker should be discharged regularly to reduce the high chlorine content. The waste generated from the chlorine discharging process is referred to as chlorine bypass dust (CBD) [21,22]. CBD mainly comprises K, Cl, and Ca, and its landfill disposal is costly. A previous study reported that K and Cl ions in the form of KCl (potassium chloride) and Ca in the form of Ca(OH)_2_ exist in CBD [23,24].

We prepared a pure KCl solution from CBD, by dissolution with tap water and filter separation. Reports on the use of reagent-grade KCl have demonstrated that it can accelerate the setting time and initial compressive strength of cement by increasing the initial hydration heat [25]. Because KCl has high alkalinity, it can be used as an alkali activator for cementitious materials like NaOH and KOH. We hypothesized that if the separated KCl were well prepared, it could show the almost same performance as reagent KCl.

This study therefore aimed to investigate the hydration characteristics of KCl separated from CBD in cementitious materials, such as ground granulated blast furnace slag and fly ash. The properties of cement-free mortar and cement-containing mortar were analyzed. The reaction characteristics between the by-products and KCl were analyzed, and the resulting strength was compared with samples prepared using other alkali activators, such as NaOH and KOH. Because a high Cl content can lead to the corrosion of the steel reinforcements in reinforced concrete, this research can be applied to plain concrete structures without steel reinforcement, such as soil solidification agents. This research can also be applied very limitedly to reinforced concrete.

## 2. Materials and Methods

### 2.1. Experimental Design

This study investigated several binders, replacement ratios, alkali activator types, and alkali activator dosages (Table 1). Ordinary Portland Cement (OPC) was used as a reference binder, and was compared to BFS and FA as representative latent hydraulic industrial by-products. The replacement ratios of BFS and FA were 0%, 50%, and 100%. The replacement ratios 0% and 50% were set using the activity measurement method per ASTM C989:2018 (Standard Specification for Ground Granulated Blast-Furnace Slag and Mortar) and the 100% replacement ratio was selected for the purpose of zero cement testing [26]. The effects of the various alkali activators, namely NaOH, KOH, and liquid separated KCl, were evaluated according to the presence or absence of cement. The content of each alkali activator was 10% of the mass of BFS and FA. The concentration of the liquid-separated KCl was determined based on precipitation and drying. The cement-free and cement-containing mortar samples (Table 2) were subjected to water curing and autoclave curing at 180 °C. The mix proportions were calculated based on a water-to-binder ratio of 40% and air content of 5%. The setting time of the paste was measured, and the flow and compressive strength of the mortar were evaluated over an aging period of 28 d.

### 2.2. Materials

Although the high content of KCl and Ca(OH)_2_ in CBD is highly soluble in water, they have different solubilities. To get the pure KCl, we mixed CBD with tap water and reacted with CO_2_ in a carbonation chamber. Ca(OH)_2_ was converted into solid calcium carbonate(CaCO_3_) easily. The solid materials, including solid calcium carbonate, were separated from the solution using a glass microfiber filter. From this process, it was possible to get the pure KCl solution.

OPC was obtained from company H in Danyang-gun of the Republic of Korea, BFS (type 3) was obtained from company S in Dangjin-si of the Republic of Korea, and FA (type 2) was obtained from power generation company E in Donghae-si of the Republic of Korea. The fineness of the OPC, BFS, and FA powders was 382, 436, and 305 m^2^/kg, respectively, and was measured by ASTM C204:2018 (Standard Test Methods for Fineness of Hydraulic Cement by Air-Permeability Apparatus) [27]. The chemical compositions of the binders and the CBD are given in Table 3. A polycarboxylate-based superplasticizer from s company in Asan-si of the Republic of Korea was used, and a superplasticizer diluted at a concentration of 20% was used in the test. Reagent-grade NaOH with a purity of >98% from company D in Ansan-si of the Republic of Korea and KOH with a purity of >85% from company D in Ansan-si of the Republic of Korea were also used as activators to serve as a comparison with the KCl separated from the CBD. The raw CBD contained CaO as well as Ca ions, while the extracted KCl mostly contained KCl minerals (Figure 1). In CBD, KCl is separated into highly soluble components, but the pH was raised using NaOH to prevent dissolution of Ca and other components. Selective separation of KCl was achieved by mixing the CBD with water at a solid-water ratio of 400 g/L at 20 °C using an INTLLAB MS-500 magnetic stirrer (INTLLAB, Weihai, China) running at 300 rpm for 10 min. The solution and solids were separated using a grade C glass microfiber (GF/C) filter with diameter 4.7 cm, pore size 1.2 µm, CAT no. 1822-047, Whatman. Separation was performed by gravity filtration. The pH of the solution was 12.7, the KCl purity was >95%, and the KCl concentration was 193 g/L. These measurements were used to calculate the solid. The main ions in the solution were K and Cl ions (>95%), while some Ca ions were also included.

### 2.3. Methodology

The setting time of the sample pastes was determined three time at a temperature of 20 ± 2 °C and a relative humidity of 60 ± 5% in accordance with the ASTM C191:2019 (Standard Test Methods for Time of Setting of Hydraulic Cement by Vicat Needle) [28]. The amount of water required to obtain a normal consistency in the fixed state was determined based on the content of each alkali activator. The alkali activator was dissolved in the sample paste and the flow of mortar was determined three times using a specified flow table based on ASTM C1437:2020—the standard test method for flow of hydraulic cement mortar [29]. The table was dropped into the paste 25 times for 15 s, and the diameter was measured.

The compressive strength of the molded mortar samples was measured five times in accordance with the ASTM C109:2020 (Standard Test Method for Compressive Strength of Hydraulic Cement Mortar) [30]. The samples used for the evaluation of compressive strength and hydration were aged 3, 7, and 28 days under water curing (3D-W, 7D-W, 28D-W, respectively) and autoclave curing (Autoclave). In particular, the water-cured samples were held at a constant temperature of 20 ± 1 °C and humidity of 60 ± 5% for 24 h, demolded, and subjected to water curing in a water tank at 20 °C. The autoclave cured samples were also held at a constant temperature of 20 ± 1 °C and a humidity of 60 ± 5% for 24 h, and subsequently cured at 180 °C and 10 atm for 8 h. The composition of the hydrates after aging for 28 d was determined two times for both sets of cured samples using a MiniFlex600 X-ray diffraction (XRD, Rigaku, Tokyo, Japan) device with a step size of 0.020° in the range between 5° and 60° 2-theta at a scanning speed of 6°/min. Cu K (λ = 1.5056 Å) was used as the Radiation source, the X-ray generator was 600 W, and the monochromator was not used. Samples used for the X-ray diffraction were kept under 45 µm by grinding.

## 3. Results

### 3.1. Mortar Flow

The flow of the cement-free and cement-containing mortar samples with NaOH, KOH, and the cement-derived KCl as activators is illustrated in Figure 2. Each activator was added to the mixing water and used in the dissolved state. Therefore, the liquid phase increased as a sum of the mixing water and activators. The fluidity of the mortar was not affected by the activators, and all the mixtures fell within the 185 ± 15 mm range due to the use of a superplasticizer. FA was found to satisfy the target fluidity even with a small amount of superplasticizer due to its spherical particles and low fineness. On the other hand, as BFS has a rough granular shape and a relatively high fineness because of its process of manufacture by grinding, a large amount of superplasticizer was used to achieve the target fluidity. There was no difference in fluidity, despite the increases in the amount of liquid phase that could affect the flow of mortar due to the replacement of mixing water and binder with the added activators. This was attributed to an increase in the viscosity of the mixing water due to the solid to liquid dissolution of the activator.

### 3.2. Paste Setting Time

The setting times of the cement-free and cement-containing mortar pastes with KCl separated from the activators CBD, NaOH, and KOH were evaluated at an activator content of 10% compared to the mass of mineral admixture (Figure 3). These samples were also used in the mortar test. Among the 100% mineral admixtures, the BFS mixtures with NaOH and KOH exhibited a final setting time of less than 3 h, which was faster than the 100% OPC sample. However, the mixture with KCl exhibited a final setting time of 15 h 37 min. The FA mixture with NaOH did not set within 24 h, while the FA mixtures with KOH and KCl set within 8 h 53 min and 17 h 17 min, respectively. The addition of KCl to both non-cement mixtures (100% BFS or FA) led to a setting time of >10 h due to the properties of the activator. However, it was encouraging that KCl led to the same level of acceleration in FA, while KOH and NaOH exhibited a significantly lower acceleration effect in FA, despite their high acceleration effect in BFSs with high hydration activity due to their latent hydraulic properties. In particular, the non-cement mixture containing NaOH in FA did not set within 16 h. These findings demonstrated that KCl had a superior effect as an alkali activator.

The use of NaOH and KOH in the 50% OPC and 50% mineral admixture binders led to rapid setting within ~1 h, regardless of the binder composition. The addition of KCl to the FA mixture led to a final setting time of 5 h 37 min, which was slightly longer than that of 100% OPC. However, the addition of KCl to the BFS mixture led to faster setting (4 h 10 min) than 100% OPC. This indicated that KCl also accelerated setting by stimulating the cement alite and interstitial minerals due to the high alkalinity (pH > 12) and salinity, despite the minimal initial stimulation due to the relatively low pH compared to NaOH and KOH. This confirmed that KCl can be used as an activator for mixtures of OPC and mineral admixture.

### 3.3. Compressive Strength of Mortar

The compressive strength of the 100% OPC mortar sample without an activator and that of the 100% BFS mortar samples with NaOH, KOH, and cement-derived KCl as activators were evaluated according to curing conditions and age (Figure 4). Further, the strength development rate of each sample compared to the strength of the OPC 100% sample was calculated as the hydration activity index. The samples with 100% FA were excluded from further investigation because they could not develop an early-stage strength of 1 MPa or higher before 28 d, despite the use of alkali activators. This lack of early-stage strength was attributed to a lack of hydration reaction activation within the samples by the alkali activators. A large amount of the FA particles was inert because the Fe_2_O_3_ content was over 11% and the 45-µm sieve residue was less than 40% according to ASTM C618:2019 due to the fineness of 305 m^2^/kg.

The 100% BFS mortar samples developed to different strengths depending on activator type. In particular, NaOH and KOH activators led to compressive strengths of >30 MPa after aging for 28 d. Although this was lower than the compressive strength of 100% OPC at 28 d (46 MPa), the non-cement mixtures based on the latent hydraulic binder exhibited a hydration activity index of >65%, which demonstrated the high possibility of using the binder alone. The samples subjected to autoclave curing exhibited higher hydration activity indices than the water-cured samples. In particular, the 100% BFSs sample with NaOH as an activator exhibited a high hydration activity index of >130% after autoclave-curing compared to OPC. Therefore, BFS is expected to achieve higher performance than OPC when used in precast concrete product manufacturing based on hydrothermal synthesis. However, the 100% BFS sample with cement-derived KCl as an activator did not exhibit an equivalent performance to the NaOH- and KOH- activated samples. Although strength development in a non-cement sample was achieved using the KCl activator, the strength was only half that of OPC. This corresponded to the setting behavior (Figure 3), where the strength development contribution of KCl via alkali stimulation was low compared to NaOH and KOH.

The compressive strength of the 100% OPC mortar sample without an activator was compared to that of the 50% mineral admixture (BFS or FA) and 50% OPC mortar samples with alkali activators, and the hydration activity indices were calculated (Figure 5). The 100% FA samples did not develop an early-stage strength of 1 MPa before 28 d, but the 50% OPC and 50% FA mixture strengthened sufficiently. In particular, the 50% FA and 50% OPC sample without an activator exhibited a hydration activity index of >50% compared to OPC at all ages, where the activity reached 60% as the age increased. This is judged to be because OPC served as an active material and contributed to the pozzolan reaction by stimulating the FA surface, where the effect increased with age. Further, it appeared that FA partially contributed to strength development due to its high amorphous Si content, thereby generating tobermorite via hydrothermal synthesis during autoclave curing. Therefore, the strength ratios of the water-cured FA samples at 28 d of age compared to the BFS samples were only 69%, 69%, 51%, and 67% for no activator, NaOH, KOH, and KCl, respectively, but increased to 81%, 110%, 75%, and 88%, respectively, under autoclave curing.

The samples comprising 100% mineral admixture did not exhibit reactivity due to their latent hydraulic properties in the absence of an activator. However, mixing of the mineral admixture with OPC led to strength development due to stimulation by CH generation during OPC hydration. Thus, the samples without an added activator also exhibited a hydration activity index of >50%. The 50% BFS and 50% OPC sample exhibited a hydration activity index of more than 100% compared to OPC at 28 days of age. However, the 50% BFS and 50% OPC samples with NaOH and KOH as activators exhibited lower strengths because OPC theoretically generates approximately 20% Ca(OH)_2_ with sufficient hydration reaction with water, and the addition of the activator adversely affected strength development due to excessive alkali [31]. Further, the minerals that were generated by the hydration of OPC at a high pH, such as ettringite, were unstable despite the strength contribution of NaOH and KOH via the stimulation of mineral admixture reactivity [32,33]. The 50% BFS and 50% OPC samples with KCl as an activator exhibited the opposite trend to the 100% BFS sample, and achieved a higher strength than those of the samples activated with NaOH and KOH, as well as the 100% OPC sample. In particular, the hydration activity index of the 50% BFS and 50% OPC sample activated with KCl exceeded 100% at all ages and reached 127% at 28 d. This demonstrated that KCl did not affect the instability of the minerals in OPC because the pH was not as high as NaOH and KOH. Cl participated in a rapid hardening reaction with OPC, where the subsequent hydration reaction of alkali C_3_A also affected BFS and FA [34,35,36]. The amounts of NaOH and KOH added in this study were excessive, while KCl was used at a level that can develop appropriate performance for the mixtures of OPC and mineral admixture. Therefore, further research on this is required to determine the optimal activator dosage for strength development.

The 50% mineral admixture (BFS or FA) and 50% OPC mortar samples with no added activator were compared to the samples with added activators to evaluate the strength intensification rate for each activator according to age and curing condition (Figure 6). The samples activated using NaOH and KOH exhibited a significant reduction in strength at all ages, while those with KCl showed improved strength. In particular, the 50% BFS and 50% OPC sample with a KCl activator exhibited a compressive strength of 34.6 MPa after 3 d, which was >90% higher than the compressive strength of the sample without an activator (18.2 MPa). This compressive strength was even higher than that of the 100% OPC sample (30.3 MPa). These findings confirmed that the use of KCl had a beneficial effect on the strength intensification rates of the samples with BFS and FA at all ages. This effect was maximized in the early stages because the addition of KCl to the 50% BFS and 50% OPC mixture facilitated a faster setting rate than the 100% OPC mixture. Fast setting time indicates that a large number of hydrates were produced throughout the sample in the initial stage, which would have filled the pores in the sample and caused high strength to be expressed. It is judged that KCl promotes the generation of hydrates that should be expressed in the long term, resulting in high initial strength. This increase in strength was attributed to accelerated hydration in the early stages, which even affected the strength at 28 d.

The compressive strength of the FA samples was generally lower than the strengths of the BFS samples. In this study, the mineral admixtures were used at the same weight ratio under the same conditions. However, ASTM C989:2018 (Standard Specification for Ground Granulated Blast-Furnace Slag and Mortar) and ASTM C618:2019 (Standard Specification for Coal Fly Ash and Raw or Calcined Natural Pozzolan for Use in Concrete) suggest mixing proportions of BFS and FA to OPC of 50% and 25%, respectively, for the measurement of the hydration activity index [37]. Overall, the use of proper mixing proportions and their effects were not negligible. Therefore, the recommended mixing proportions of mineral admixtures will be considered in future studies. Impact evaluation for the use of cement-derived KCl as an activator should also be further investigated.

### 3.4. Mineral Composition

Mineral analysis was conducted for the 50% BFS and 50% OPC (Figure 7) and 50% FA and 50% OPC samples (Figure 8). Among the BFS mixtures, the water-cured sample activated with KCl contained hydrocalumite, which was not observed in the samples that used the other activators. Hydrocalumite is known as Friedel’s salt [38]. The presence of Friedel’s salt can compromise the strength of a 100% OPC sample, but this adverse effect is minimized by the presence of the mineral admixture [39]. In addition, a previous study reported that the generation of Friedel’s salt in a mixture of OPC with a substantial proportion of BFS led to enhanced strength due to the filling of pores [40]. Thus, the increased strength due to the use of KCl in this study was also likely attributed the generation of Friedel’s salt. The Friedel’s salt-based hydrate production itself did not appear to occupy the pores significantly compared to the overall hydrate, but the triggering reaction of Friedel’s salt might have influenced the promotion of amorphous products such as the C–H–S system. The hydrate generated by this facilitation reaction filled the voids of the test sample and contributed to the increase in strength. Water curing led to the formation of hydrocalumite, while autoclave curing did not. This indicated that autoclave curing at 180 °C led to the decomposition of CaClOH that constitutes hydrocalumite at ~100 °C [41]. Thus, the loss of strength due to autoclave curing of the BFS mixtures was related to the decomposition of hydrocalumite and amorphous minerals. Friedel’s salt-based hydrate was decomposed in all test samples, but it seems that the BFS mixture, where a relatively substantial number of Friedel’s salt-based hydrates are present, was significantly affected. In the case of the FA mixture, the production of tobermorite under high-temperature and high-pressure conditions seems to have had a greater influence on the intensity than the decomposition of the product.

The FA mixtures activated with KCl also produced hydrocalumite under water curing, but not autoclave curing. In addition to the decomposition of hydrocalumite, tobermorite and katoite were formed during autoclave curing and contributed to strength development with low density compared to C–S–H generated at room temperature, and had a high volume stability. The positive pore-filling effect of the minerals generated in the FA mixtures was larger than the negative effect of the decomposition of minerals such as hydrocalumite. Thus, the compressive strength increased overall.

## 4. Conclusions

This study provides basic insights into the use of cement-derived KCl as an activator for plain mortar and concrete. Reagent-grade KOH and NaOH were used as alkali activators for comparison, and the basic characteristics of mortar comprising only mineral admixture or mineral admixture with OPC as a binder were analyzed. The conclusions of this study are as follows:The setting of the mortar paste due to the addition of KCl was slightly slower than pastes containing the strong alkali activators. However, setting within 24 h was achieved when the mineral admixture was used alone, and the setting of the mineral admixture and OPC mixtures was accelerated.The fluidity of the mortar was maintained or decreased slightly with the addition of KCl and the alkali activators due to increased mortar viscosity.Analysis of compressive strength demonstrated that the 100% mineral admixture samples activated with KCl were not as strong as the samples activated with NaOH and KOH. However, the activity index values of the 50% BFS and 50% OPC sample activated with KCl was 100% or more at all ages, and especially 127% at 28 d and 135% under autoclave condition.Water curing of the samples with KCl led to the formation of hydrocalumite, or Friedel’s salt. However, hydrocalumite decomposed during autoclave curing at 180 °C.The addition of KCl accelerated setting, improved the early stage hydration activity of the mineral admixtures and OPC mixtures, and increased the strength by >20% compared to the mixtures without an activator at 28 d.

Overall, these findings demonstrated that KCl has higher applicability in mixtures of cement and mineral admixtures than in mineral admixtures alone. The optimal application conditions of KCl as an activator for plain concrete products should be further investigated based on the analysis of its properties in low-cement mixtures at different proportions, and in the presence of hydrocalumite based on a controlled accelerated curing temperature.

## Figures and Tables

**Figure 1 materials-14-06091-f001:**
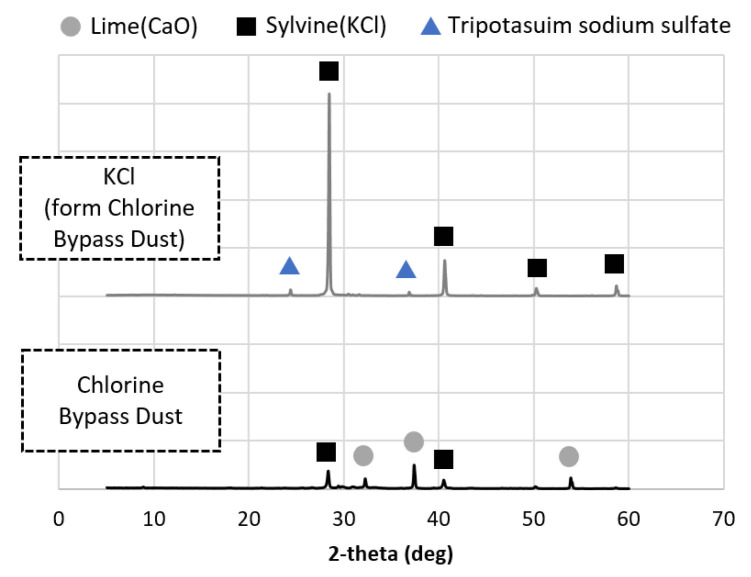
Mineral composition of CBD and KCl.

**Figure 2 materials-14-06091-f002:**
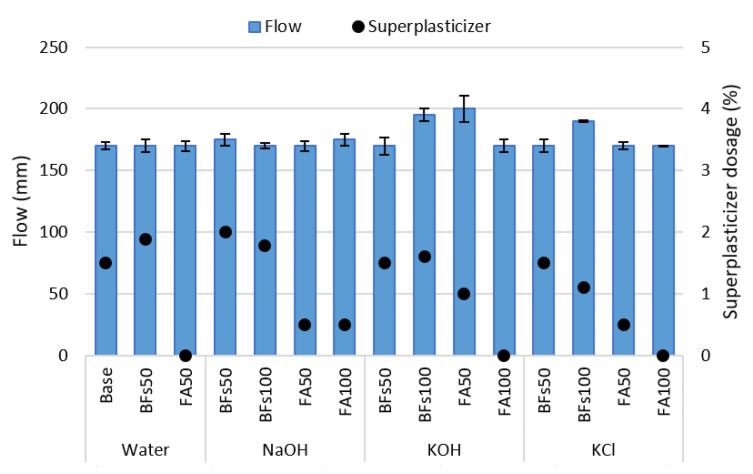
Flow of samples with KCl separated from CBD, NaOH and KOH as activators.

**Figure 3 materials-14-06091-f003:**
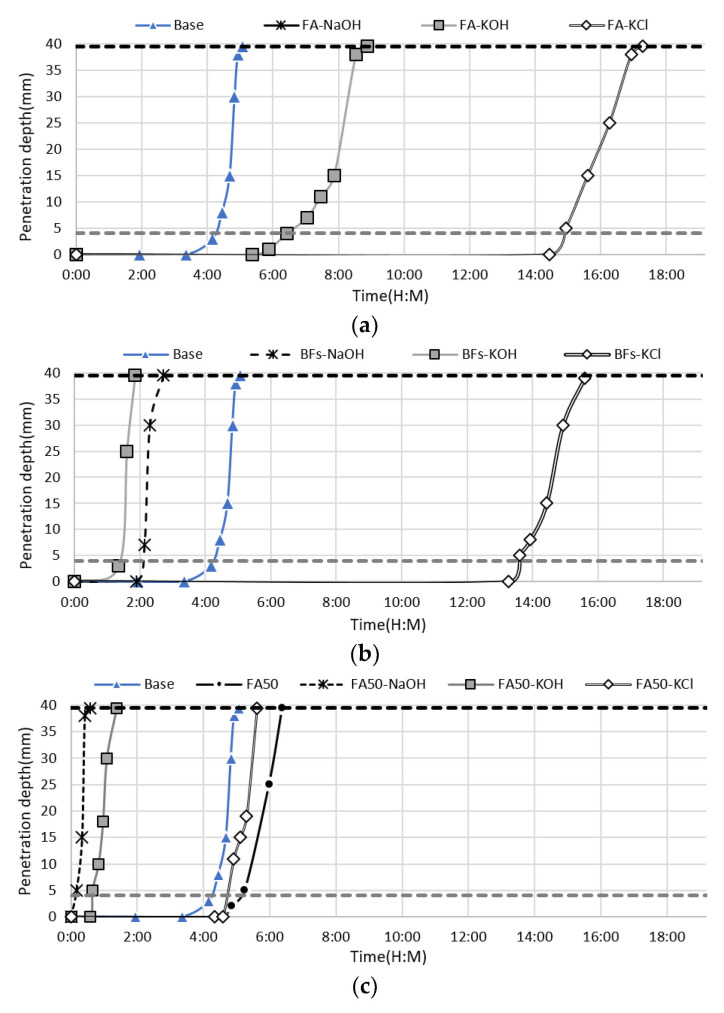
Setting of the cement-containing and non-cement mortar with KCl separated from CBD, NaOH, and KOH as activators, namely, (**a**) 100% FA, (**b**) 100% BFS, (**c**) 50% FA and 50% OPC, and (**d**) 50% BFS and 50% OPC.

**Figure 4 materials-14-06091-f004:**
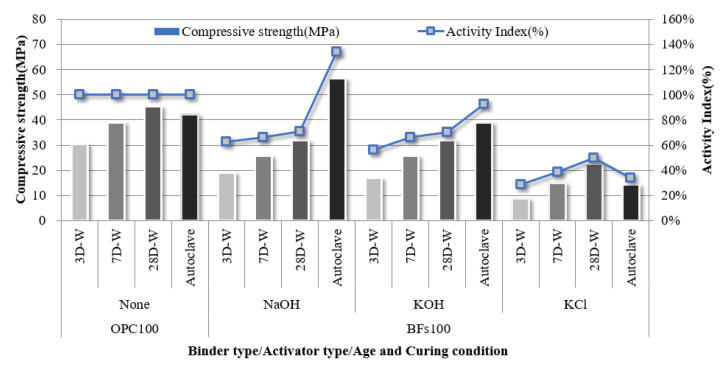
Compressive strength and activity index of the 100% BFS mortar sample with KCl separated from CBD, NaOH, and KOH as activators.

**Figure 5 materials-14-06091-f005:**
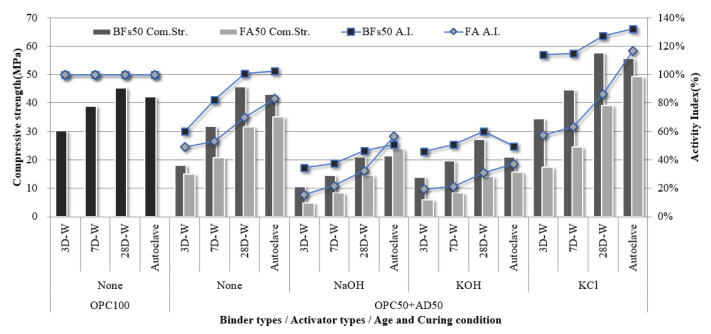
Compressive strength and activity index of the 50% mineral admixture (BFS or FA) and 50% OPC mortar samples with KCl separated from CBD, NaOH, and KOH as activators.

**Figure 6 materials-14-06091-f006:**
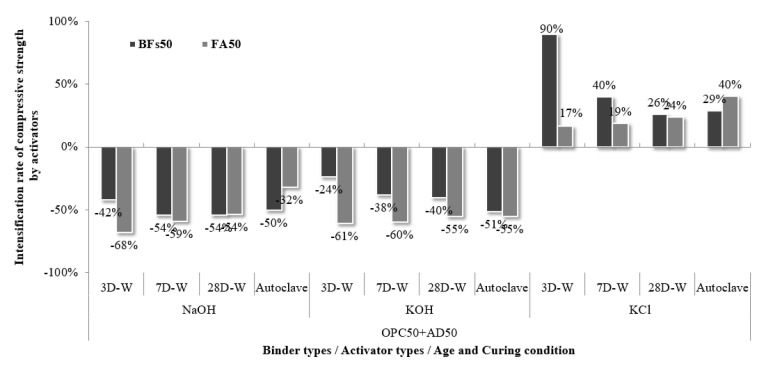
Compressive strength intensification rate of the KCl separated from CBD, NaOH, and KOH activators in the 50% mineral admixture (BFS or FA) and 50% OPC mortar samples.

**Figure 7 materials-14-06091-f007:**
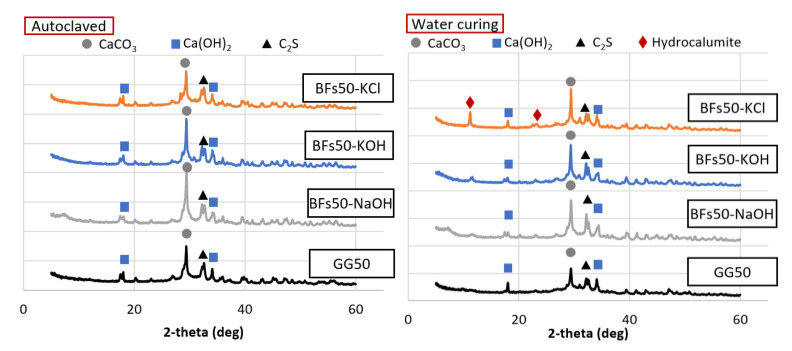
Mineral analysis of the 50% BFS and 50% OPC mortar samples with KCl separated from CBD, NaOH, and KOH as activators.

**Figure 8 materials-14-06091-f008:**
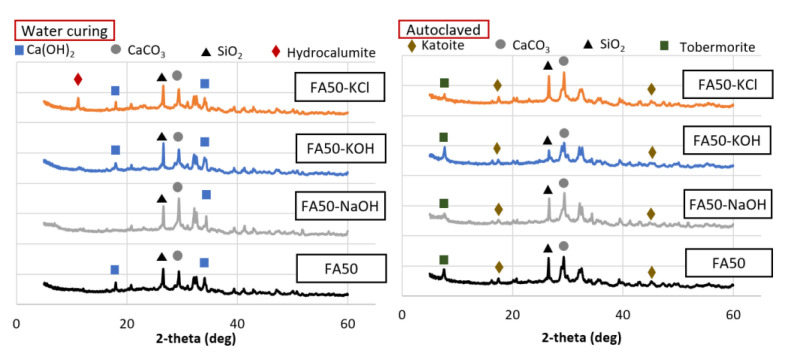
Mineral analysis of the 50% FA and 50% OPC mortar samples with KCl separated from CBD, NaOH, and KOH as activators.

**Table 1 materials-14-06091-t001:** Binders, replacement ratios, alkali activator types, and alkali activator dosages used in this experimental design.

Binder Type	Mineral Admixture Replacement Ratio (wt%)	Alkali Activator Type	Alkali Activator Dosage (wt%)
OPC BFS FA	0 50 100	None NaOH KOH KCl (by-product)	10% to mineral admixture

**Table 2 materials-14-06091-t002:** Mixtures used to prepare cement-containing and non-cement mortar samples.

Mixture		W/B (%)	Air (%)	Water (kg/m^3^)	Binder (kg/m^3^)			Sand (kg/m^3^)	Alkaline Activator (kg/m^3^)			Total (kg/m^3^)
					OPC	BFS	FA		NaOH	KOH	KCl	
None	OPC100	40	5	206	515	-	-	1545	-	-	-	2265
	BFS50			204	255	255	-	1532	-	-	-	2247
	FA50			201	251	-	251	1505	-	-	-	2208
NaOH	BFS50			204	255	255	-	1532	26	-	-	2273
	BFS100			203	-	507	-	1520	51	-	-	2280
	FA50			201	251	-	251	1505	25	-	-	2233
	FA100			196	-	-	489	1468	49	-	-	2202
KOH	BFS50			204	255	255	-	1532	-	26	-	2273
	BFS100			203	-	507	-	1520	-	51	-	2280
	FA50			201	251	-	251	1505	-	25	-	2233
	FA100			196	-	-	489	1468	-	49	-	2202
KCl	BFS50			204	255	255	-	1532	-	-	26	2273
	BFS100			203	-	507	-	1520	-	-	51	2280
	FA50			201	251	-	251	1505	-	-	25	2233
	FA100			196	-	-	489	1468	-	-	49	2202

**Table 3 materials-14-06091-t003:** Chemical composition of the binders and the CBD used in this experiment.

	CaO	SiO_2_	Al_2_O_3_	Fe_2_O_3_	MgO	SO_3_	K_2_O	Cl	Other	Total
OPC	65.84	17.44	3.84	3.27	3.20	3.25	1.44	0.05	1.67	100
BFS	48.13	30.51	12.93	0.55	2.70	2.58	0.60	0.01	1.99	100
FA	6.91	49.70	21.66	11.13	2.45	0.87	1.53	0.19	5.56	100
CBD	44.4	7.74	3.23	1.92	0.55	13.68	9.87	15.7	2.91	100

## Data Availability

Not applicable.

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
