# Peer review of "KCl Extracted from Chlorine Bypass Dust as Activator for Plain Concrete"

_materials, 2021, doi:10.3390/ma14206091_

Round 1
Reviewer 1 Report
The abstract is well written as it considers obtaining a general overview of the significant finding. However, the short problem statement related to the current study needs to be highlighted. The author failed to deliver a comprehensive literature review on chlorine bypass dust (CDB) and KCI. The methodology is appropriate in the current work but detailed information on the characterization technique is required.
The following suggestion and comments should be taken
- What is KCI?
- Introduction. Paragraph 1. Please remove seem it was not reflected to the current work.
- Section 2.1. Based on what standard or information that the experimental design has been followed?
- Table 1. Why BBFS 50 wt% and FA 100 wt%?
- “…3818,4356, 3051 cm2/g”. Please revised the fineness unit of the materials.
- Please define more on the selective separation of KCI.
- Figure 1. What type of analysis was used to determine the lime and sylvine (KCI)?
- Line 141-143. The required amount of superplasticizer is majorly influenced by the type of chemical composition consist in the material. Please revise in detail.
- What did the author mean by dissolution activator? Or it was preferable written as dissolution aluminosilicate?
- Figure 2. Please add standard deviation in the image. Also, please replace a higher-quality image of the result throughout the manuscript.
- Line 173-174. The addition of KCI in the system is still unclear as the component is missing in Table 2.
- “…the generation of sufficient CH…” How the author did confirm? This was needed to be supported with the XRD result.
- Please describe in detail the latent hydraulic property.
- “…generates approximately 20% to 25% Ca(OH)2…” Did the author analyze the production of Ca(OH)2?
- Line 269-270. Please extend the discussion on faster setting rate towards hydration strength intensification rate.
- The discussion and analysis of the result required for correlation between results. And please add more CRITICAL DISCUSSION in the Results and Discussion.
- Please check the overall format of the references and make sure all of it been standardized.
Author Response
Point 1 : What is KCI?
Response 1 : KCl refers to Potassium Chloride, and although it does not have a pH as high as an alkali-stimulating agent, the pH of KCl used in the experiment is 12 or higher.
Point 2 : Introduction. Paragraph 1. Please remove seem it was not reflected to the current work.
Response 2: We revised it based on the reviewer's comments.
Point 3 : Section 2.1. Based on what standard or information that the experimental design has been followed?
Response 3: The replacement ratios of BFS and FA were 0%, 50%, and 100%. The replacement ratios 0% and 50% were set using the activity measurement method per ASTM C989:2018 (Standard Specification for Ground Granulated Blast-Furnace Slag and Mortar) and the 100% replacement ratio was selected for the purpose of zero cement testing [26]. The effects of the various alkali activators, namely NaOH, KOH, and liquid separated KCl, were evaluated according to the presence or absence of cement.
Point 4 : Table 1. Why BBFS 50 wt% and FA 100 wt%?
Response 4 : We conducted the experiment by replacing 50% and 100% of the FA and BFS with cement. However, when FA is used 100%, it had no strength, so there is no test data. I revised the table.
Point 5 : “…3818,4356, 3051 cm2/g”. Please revised the fineness unit of the materials.
Response 5: We revised it based on the reviewer's comments.
Point 6 : Please define more on the selective separation of KCI.
Response 6 :We added the contents based on the reviewer's comments.
Point 7 : Figure 1. What type of analysis was used to determine the lime and sylvine (KCI)?
Response 7 : The composition of the hydrates after aging for 28 d was determined two times for both sets of cured samples using a Rigaku MiniFlex600 X-ray diffraction (XRD) device with a step size of 0.020° in the range between 5° and 60° 2-theta at a scanning speed of 6°/min.
Point 8 : Line 141-143. The required amount of superplasticizer is majorly influenced by the type of chemical composition consist in the material. Please revise in detail.
Response 8 : If a material with a similar granular shape to fineness was used, the cause of the difference in fluidity could be said to be a chemical factor. However, in this experiment, the physical factor seems to be greater because the granular shape and fineness of the material are different. The contents of these parts were added and supplemented.
Point 9 : What did the author mean by dissolution activator? Or it was preferable written as dissolution aluminosilicate?
Response 9: The dissolution of the activator in the fluid part is when a solid activator is dissolved in a liquid state.
Point 10 : Figure 2. Please add standard deviation in the image. Also, please replace a higher-quality image of the result throughout the manuscript.
Response 10: We revised it based on the reviewer's comments.
Point 11 : Line 173-174. The addition of KCI in the system is still unclear as the component is missing in Table 2.
Response 11 : We revised it based on the reviewer's comments.
Point 12 : “…the generation of sufficient CH…” How the author did confirm? This was needed to be supported with the XRD result.
Response 12 : When CH occurs, it is difficult to check because it disappears in response to FA and it was a presumed phenomenon. We revised it based on the reviewer's comments.
Point 13 : Please describe in detail the latent hydraulic property.
Response 13 : The sample using KCl (potassium chloride) as an alkali activator conducted in this experiment shows initial fast setting time and high strength. With these properties, it is judged that concrete can be made from zero cement and low cement, but long-term strength expression is expected to be slower than that of test subjects that do not use KCl.
Point 14 : “…generates approximately 20% to 25% Ca(OH)2…” Did the author analyze the production of Ca(OH)2?
Response 14 : The contents were not measured; however, they were referred to in the literature. We added the reference literature.
Point 15 : Line 269-270. Please extend the discussion on faster setting rate towards hydration strength intensification rate.
Response 15: We revised it based on the reviewer's comments.
Point 16 : The discussion and analysis of the result required for correlation between results. And please add more CRITICAL DISCUSSION in the Results and Discussion.
Response 16: We revised it based on the reviewer's comments.
Point 17 : Please check the overall format of the references and make sure all of it been standardized.
Response 17: We revised it based on the reviewer's comments.

Reviewer 2 Report
The subject of the article is “KCl extracted from chlorine bypass dust as activator for plain concrete”
It is suggested that the following additions be made in order to improve the article.
1.The difference of the article from other articles and its contribution to science should be written.
- For a better understanding of the results obtained in the research, it would be good to perform statistical analysis.
Performing ANOVA analysis for the data obtained in the study (replacement ratios, alkali activator types, and alkali activator dosages) will scientifically reveal the differences between the obtained data.

Author Response
-
Point 1 : The difference of the article from other articles and its contribution to science should be written.
Response 1 : We revised it based on the reviewer's comments.
Point 2 : For a better understanding of the results obtained in the research, it would be good to perform statistical analysis.
Response 2 : We will try to supplement statistical analysis through further research.
Performing ANOVA analysis for the data obtained in the study (replacement ratios, alkali activator types, and alkali activator dosages) will scientifically reveal the differences between the obtained data.

Reviewer 3 Report
Excellent innovation and experimental methods are advanced, and English statements are used correctly, but it is necessary to make a closer to some details.

Author Response
Point 1 : Excellent innovation and experimental methods are advanced, and English statements are used correctly, but it is necessary to make a closer to some details.
Response 1: We revised it based on the reviewer's comments.
Reviewer 4 Report
Comments on the article titled “KCl extracted from chlorine bypass dust as activator for plain concrete” by Hongbeom Choi, Jinman Kim, Sunmi Choi and Sungsu Kim submitted to MDPI Materials
This work is very interesting and valuable however before publication some issues should be clarified.
My main concern is the compliance of the manuscript text with the title of the work. The title suggests that it concerns mainly the usage of KCl extracted from chlorine bypass dust (CBD). It seems that the workload has been shifted to checking different binders, replacement ratios, comparing the types of alkali activators. The real purpose was to reduce the amount of cement used in concrete by replacing it with FA or BFS. Results were compared with usage of Ordinary Portland Cement. This issue should be clarified and usage of KCl indicated.In any case the detailed method of KCl extraction should be inserted in Materials and Methods. The procedure for KCL separation should be clear enough to repeat it independently by other researchers.
-what an alkali solvent means (line105)?
-specify rot/min for stirring, specify the time of stirring and a temperature during this process; please add the producer of magnetic stirrer (type, name, name of producer, country of producer)- give the producer (name and country) of grade C glass microfiber (GF/C) filter with a detailed specification (diameter, flow speed (mL/min), pore size, etc). Was the separation gravimetric or the liquid was pumped into filter? Was the filter sterile or non-sterile. These are very important details.
- how the KCl purity and concentration were evaluated?
Which component is indicated by a X-ray diffraction pattern with 2-theta c.a.24deg and which at 37 deg on Fig 1? It would be more convenient to change the legends of figures 1 and 8 to X-ray diffraction pattern.........etc. Besides it is necessary to include the conditions of measuremants of minerals (the apparatus type, kind of power supply, type of goniometer, kind of a cathode used, lamp voltage, accuracy of measurement data etc.)
Please give names of producers, cities and countries for company H, S and E. Give the names of producers and countries for reagents (reagent-grade KOH and NaOH).
flexural strength? Was it really measured? Line 89? Remove!
Figures general – introduce spaces the measured quantity and its unit
Figure 1 ( as commented above) Which component is indicated by a X-ray diffraction pattern with 2-theta c.a.24deg and which at 37 deg
Figure 2 There is no information about the kind of superplasticizer used. Please add this information to Materials and Methods
Figure 3 unify sample designations in tables and figures (example Bfs50 and BFS50)Figure 4,5 and 6 include samples designations into Materials and Methods (3D-W, ....28D-W, etc)
Other technical corrections
1. Please pay attention to the font size in affiliations (capital letters in name of the Institution) line 52. Introduction: Please remove lines 24-32 as this is the description how the Abstract should be written3. Please insert spaces before citing in the brackets throughout the text4. Change the layouts of Tables, make sure that they are readable for the reader (bracket in units in Table 2 are shifted)5. the descriptions in the figures are not sharp, one can see the edges of the inserted/glued sections. Please make sure that the resolution of the images is high enough 6.The unit oC appears in text in different formats. Please pay attention to the font used throughout the text.7. Is the term “ground granulated blast furnace slag” correct? Was it ground or granulated? granulate was grinded? How? Give the details8. References should be corrected according to the requirements of Materials –MDPI Journal. (sometimes full names as well as abbreviations for journal names are used; in several quotations first and last names are in the wrong order; missing spaces; sometimes dots are present after abbreviation of a first name appear sometimes not)9. Language: please let the native speaker or a colleague who is fluent in English to read and correct the text. Think about using word “sample” instead of a “specimen” etc.
Author Response
Point 1 :what an alkali solvent means (line105)?
Response 1: It means solid-water ratio. We revised it based on the reviewer's comments.
Point 2 :specify rot/min for stirring, specify the time of stirring and a temperature during this process; please add the producer of magnetic stirrer (type, name, name of producer, country of producer)- give the producer (name and country) of grade C glass microfiber (GF/C) filter with a detailed specification (diameter, flow speed (mL/min), pore size, etc). Was the separation gravimetric or the liquid was pumped into filter? Was the filter sterile or non-sterile. These are very important details.
Response 2: We added the contents based on the reviewer's comments.
Point 3 : how the KCl purity and concentration were evaluated?
Response 3: In the case of purity, it was measured by X-ray fluorescence, and the concentration measurement was confirmed by drying the solution and measuring the mass of the solid.
Point 4 : Which component is indicated by a X-ray diffraction pattern with 2-theta c.a.24deg and which at 37 deg on Fig 1? It would be more convenient to change the legends of figures 1 and 8 to X-ray diffraction pattern.........etc. Besides it is necessary to include the conditions of measuremants of minerals (the apparatus type, kind of power supply, type of goniometer, kind of a cathode used, lamp voltage, accuracy of measurement data etc.)
Response 4: The mineral shown in KCl was measured with Tri potassium sodium sulphate. When using XRD, it was judged that it was not a major mineral because the amount was not large, so it was not displayed.
We revised it based on the reviewer's comments.
Point 5 : Please give names of producers, cities and countries for company H, S and E. Give the names of producers and countries for reagents (reagent-grade KOH and NaOH).
Response 5: We revised it based on the reviewer's comments.
Point 6 : flexural strength? Was it really measured? Line 89? Remove!
Response 6: We revised it based on the reviewer's comments.
Point 7 : Figures general – introduce spaces the measured quantity and its unit
Response 7: I added the content to the measured quantity as per the methodology in the text.
Point 8 : Figure 1 ( as commented above) Which component is indicated by a X-ray diffraction pattern with 2-theta c.a.24deg and which at 37 deg
Response 8: We revised it based on the reviewer's comments.
Point 9 : Figure 2 There is no information about the kind of superplasticizer used. Please add this information to Materials and Methods
Response 9: We revised it based on the reviewer's comments.
Point 10 : Figure 3 unify sample designations in tables and figures (example Bfs50 and BFS50)Figure 4,5 and 6 include samples designations into Materials and Methods (3D-W, ....28D-W, etc)
Response 10: We revised it based on the reviewer's comments.
Point 11 : Other technical corrections
- Please pay attention to the font size in affiliations (capital letters in name of the Institution) line 5
- Introduction: Please remove lines 24-32 as this is the description how the Abstract should be written
- Please insert spaces before citing in the brackets throughout the text
- Change the layouts of Tables, make sure that they are readable for the reader (bracket in units in Table 2 are shifted)
- the descriptions in the figures are not sharp, one can see the edges of the inserted/glued sections. Please make sure that the resolution of the images is high enough
6.The unit oC appears in text in different formats. Please pay attention to the font used throughout the text.
- Is the term “ground granulated blast furnace slag” correct? Was it ground or granulated? granulate was grinded? How? Give the details
- References should be corrected according to the requirements of Materials –MDPI Journal. (sometimes full names as well as abbreviations for journal names are used; in several quotations first and last names are in the wrong order; missing spaces; sometimes dots are present after abbreviation of a first name appear sometimes not)
- Language: please let the native speaker or a colleague who is fluent in English to read and correct the text. Think about using word “sample” instead of a “specimen” etc.
Response 11: We have revised it based on all of the above suggestions.

Reviewer 5 Report
In this paper, the effect of KCl seperated from chlorine bypass dust as an activator on properties of plain concrete was investigated through a series of experiments. The paper is well written and the methodology is clearly presented and explained. The paper shows a very detailed introduction, a well-described methodology and a discussion of results. In general, this paper is interesting and is in the journal scope. Therefore, this article is acceptable but it needs some revision:
(1) The first paragraph of the section of Introduction is the template format, which should be removed.
(2) In the section of Introduction, it is stated that the properties of cement-based materials can be improved by adding alkali activators. Some up-to-date results on this aspect and the use of KCl should be introducted, and some up-to-date literatures are suggested to be used in background introduction part in the sections of Introduction, and References List for completeness of your study and the references:
Bonding behavior of concrete matrix and alkali-activated mortar incorporating nano-SiO2 and polyvinyl alcohol fiber: Theoretical analysis and prediction model. Ceramics International. 2021, https://doi.org/10.1016/j.ceramint.2021.08.044.
Mechanical properties and prediction of fracture parameters of geopolymer/alkali-activated mortar modified with PVA fiber and nano-SiO2, Ceramics International, 2020, 46 (12): 20027-20037.
Numerical modeling of rebar-matrix bond behaviors of nano-SiO2 and PVA fiber reinforced geopolymer composites, Ceramics International, 2021, 47(8): 11727–11737.
(3) In order to enhance the convenience to reference the mix proportions, the data in Table 2 is suggested to reserve integer.
(4) The quality of the figures in the manuscript should be improved.
(5) The section of results discussion should be strengthened. Besides some quantitative analysis on the results, some mechanism analysis and discussion is needed.
(6) In order to help the readers to understand the related mechanism, some SEM photos and related analysis may be helpful.
Author Response
Point 1 : The first paragraph of the section of Introduction is the template format, which should be removed.
Response 1: We revised it based on the reviewer's comments.
Point 2 : In the section of Introduction, it is stated that the properties of cement-based materials can be improved by adding alkali activators. Some up-to-date results on this aspect and the use of KCl should be introducted, and some up-to-date literatures are suggested to be used in background introduction part in the sections of Introduction, and References List for completeness of your study and the references:
Bonding behavior of concrete matrix and alkali-activated mortar incorporating nano-SiO2 and polyvinyl alcohol fiber: Theoretical analysis and prediction model. Ceramics International. 2021, https://doi.org/10.1016/j.ceramint.2021.08.044.
Mechanical properties and prediction of fracture parameters of geopolymer/alkali-activated mortar modified with PVA fiber and nano-SiO2, Ceramics International, 2020, 46 (12): 20027-20037.
Numerical modeling of rebar-matrix bond behaviors of nano-SiO2 and PVA fiber reinforced geopolymer composites, Ceramics International, 2021, 47(8): 11727–11737.
Response 2 : We revised it based on the reviewer's comments.
Point 3 : In order to enhance the convenience to reference the mix proportions, the data in Table 2 is suggested to reserve integer.
Response 3 : We revised it based on the reviewer's comments.
Point 4 : The quality of the figures in the manuscript should be improved.
Response 4 : We revised it based on the reviewer's comments.
Point 5 : The section of results discussion should be strengthened. Besides some quantitative analysis on the results, some mechanism analysis and discussion is needed.
Response 5 : We revised it based on the reviewer's comments.
Point 6 : In order to help the readers to understand the related mechanism, some SEM photos and related analysis may be helpful.
Response 6 : Per your comment, we have to analyze the image, but it has not been performed at the moment. Through additional research, we will be able to conduct structured research including image analysis.

Round 2
Reviewer 1 Report
The author has made corrections for all highlight comments.
Reviewer 4 Report
Comments on the new ammended version of the manuscript "KCL extracted from chlorine bypass dust as activator for plain concrete" by Choi et al.
The new version of manuscript is much better written as compared to the previous one. My main concern was the section of Materials and methods which has been substantially improved.
Substantial changes:
- The authors added details concerning materials ( nemes of producers, country), reagents used (producer, name, country, purity) as well as about some procedures (temperature, time, speed of magnetic stirres etc). Thanks to this the other authors are able to repeat the experiments.
- The layout of Tables has been changed
- The quality of the Figures has been substantially improved
- The language of the manuscript has been improved (for example word specimen replaced with sample)
- The results have been better justified
- The References section has been corrected to comply with the requirements of Materials (proper order of names, yera of publication, names of Journals have been unified)
- Small spelling and editting mistakes were corrected (part of the unnecessary text from lines 23-32 in previous version
Reviewer 5 Report
The manuscript can be accepted for publication in its current version.